# Characterization of Fertility Clinic Attendees in the Abu Dhabi Emirate, United Arab Emirates: A Cross-Sectional Study

**DOI:** 10.3390/ijerph20031692

**Published:** 2023-01-17

**Authors:** Noor Motea Abdo, Hafiz Ahmad, Tom Loney, Panayota Napoleon Zarmakoupis, Irfan Aslam, Shazia Irfan, Michal Grivna, Luai A. Ahmed, Rami H. Al-Rifai

**Affiliations:** 1Institute of Public Health, College of Medicine and Health Sciences, United Arab Emirates University, Al Ain P.O. Box 17666, United Arab Emirates; 2Department of Medical Microbiology & Immunology, RAK College of Medical Sciences, RAK Medical and Health Sciences University, Ras Al Khaimah P.O. Box 11172, United Arab Emirates; 3Molecular Division, RAK Hospital, Ras al Khaimah P.O. Box 11393, United Arab Emirates; 4College of Medicine, Mohammed Bin Rashid University of Medicine and Health Sciences, Dubai P.O. Box 505055, United Arab Emirates; 5IVF-Infertility Services-MEIVIS Department, Tawam Hospital, Al Ain P.O. Box 15258, United Arab Emirates; 6HealthPlus Fertility Center, HealthPlus Network of Specialty Centers, Abu Dhabi, United Arab Emirates; 7Department of Public Health and Preventive Medicine, Second Faculty of Medicine, Charles University, V Úvalu 84, 150 06 Prague, Czech Republic; 8Zayed Center for Health Sciences, United Arab Emirates University, Al Ain P.O. Box 17666, United Arab Emirates

**Keywords:** infertility, pregnancy, reproductive health, primary infertility, conception

## Abstract

This study describes the primary and secondary infertility in patients attending fertility clinics and reports factors associated with primary infertility. A cross-sectional survey was conducted in two fertility clinics in Abu Dhabi Emirate, United Arab Emirates (UAE) between December 2020 and May 2021. The collected information covered sociodemographic, lifestyle, medical, and fertility-related characteristics. The mean age and age at marriage (±SD) of the 928 patients were 35.7 (±6.7) and 25.2 (±6.3) years, respectively. Of the total, 72.0% were obese and overweight, 26.6% reported a consanguineous marriage, and 12.5% were smokers. Secondary infertility (62.5%) was more frequent than primary infertility (37.5%). Primary infertility was inversely associated with age (aOR, 0.94, 95% CI: 0.91–0.98) and not being overweight (aOR, 0.6, 95% CI: 0.4–0.9) while positively associated with a nationality other than Middle Eastern nationality (aOR, 1.9, 95% CI: 1.1–3.3), married for ≤5 years (aOR, 6.0, 95% CI: 3.9–9.3), in a nonconsanguineous marriage (aOR, 2.4, 95% CI: 1.5–3.9), having a respiratory disease (aOR, 2.3, 95% CI: 1.1–4.6), an increased age at puberty (aOR, 1.2, 95% CI: 1.0–1.3), and self-reported 6–<12 months (aOR, 2.4, 95% CI: 1.2–5.1) and ≥12 months (aOR, 3.4, 95% CI: 1.8–6.4) infertility. Patients with primary infertility were more likely to be diagnosed with infertility of an ovulation, tubal, or uterine origin (aOR, 3.9, 95% CI: 1.9–7.9). Secondary infertility was more common than primary infertility. Several preventable fertility-related risk factors including overweight, smoking, and diabetes were found to be common among the fertility clinic attendees.

## 1. Introduction

Infertility is a disease of the reproductive system, defined as the inability to conceive after at least 12 months of regular unprotected sexual intercourse [1]. The World Health Organization (WHO) has approximated that 48 million couples are affected by infertility, with a global estimated prevalence of 10–15% [1]. Infertility can be primary or secondary. Primary infertility is defined as the complete inability to achieve pregnancy, whereas secondary infertility is defined as having at least one positive pregnancy. Both types of infertility can be attributed to female, male, couple, or unexplained infertility. Female infertility is mainly caused by menstrual and ovulation irregularities, whereas male infertility is caused primarily by abnormalities in semen parameters [1].

The prevalence of infertility among child-seeking women varies by region. In 2010, the prevalence of primary infertility was estimated to be 1.5% in Latin America and the Caribbean region and 2.6% in North Africa and the Middle East. The prevalence of secondary infertility ranged from 7.2% in high-income countries and North Africa and the Middle East to 18.0% in central and eastern Europe and central Asia [2]. The burden of infertility also varies by sex, with the age-standardized rates of female and male infertility in 195 countries estimated to be 1571.4 and 768.6 per 100,000 persons, respectively [3]. Infertility is a serious health issue with profound consequences, leading to psychological, social, and economic stresses on couples and their families [4]. Infertility is a multifactorial condition, with many factors contributing to its complex etiology. Advanced age at marriage leads to delayed childbearing and has been reported to be associated with fertility complications and unfavorable pregnancy outcomes [5]. Several medical conditions including increased age at menarche and irregular menstruation [6], genital infections [7], diabetes mellitus [8], hypothyroidism [9], and family history of infertility [10] have been studied for their influence on reproduction. Moreover, several nutritional supplements have a certain degree of effect on fertility. Regular intake of folic acid significantly reduced the risk of anovulatory or oligo-ovulatory sterility in females [11]. In males, steroid use has a prolonged effect on reproductive health [12]. Considering the differences in the anatomy and physiology of the reproductive organs between males and females, sex-specific factors have been investigated widely. In females, both menstrual cycle irregularity and age of menarche were found to significantly affect the time of pregnancy [13]. In males, varicocele is a common cause of infertility [14], while the link between caffeine intake and time to pregnancy [15], and between repeated exposure to high water temperatures (e.g., hot tubs, jacuzzis, hot baths, or saunas) or exposure to radiation, and the quality of semen is still under debate [16,17].

Increased demands for in vitro fertilization (IVF) and fertility treatments are indicators of a progressive infertility rate in a region. The current size of the IVF market in the Middle East and North Africa (MENA) is around USD 10 billion [18]. Approximately one in six couples in the United Arab Emirates (UAE) have difficulties conceiving [19], and the total fertility rate in the UAE has decreased from 2.7 births per woman in 2000 to 1.5 births per woman in 2020 [20]. Furthermore, many factors associated with infertility are common in the UAE. One such factor is increased age at marriage; in 2001, 2011, and 2018, the median ages at marriage were 24.6, 25.7, and 26.1 years, respectively, among women and 25.9, 26.7, and 29.1 years, respectively, among men [21]. Other factors associated with infertility that are common in the UAE include overweight and obesity [22], high tobacco utilization [23], consanguineous marriage [24], a lack of physical activity, and a high prevalence of chronic comorbidities, especially diabetes mellitus [25]. Few epidemiological studies to date, however, have characterized infertile populations in the UAE, and empirical data are lacking.

The present study identifies the sociodemographic, lifestyle, medical, and fertility-related characteristics of patients seeking infertility treatment in two fertility clinics in the Emirate of Abu Dhabi, UAE. This study also describes patients based on the type of infertility (primary or secondary) and identifies factors associated with primary infertility.

## 2. Materials and Methods

### 2.1. Study Design, Setting, and Study Population

This cross-sectional study included consecutive patients attending two major fertility clinics in Abu Dhabi Emirate, UAE, from December 2020 until May 2021. These two clinics are the HealthPlus Fertility and Women Centre, a private clinic in Abu Dhabi city, and the Tawam IVF Centre, a public clinic located in Tawam hospital, one of the largest government hospitals in the Al Ain region. All patients seeking infertility treatment were invited to participate, including those fulfilling the WHO definition of infertility (i.e., an inability to conceive after ≥12 months of regular unprotected sexual intercourse [1]), and the clinical practice guidelines in the UAE (women aged >35 years unable to conceive after 6 months of regular unprotected sexual intercourse [26]). Patients were eligible to participate if they were ≥18 years and seeking fertility treatment. Patients aged <18 years, spontaneously pregnant women, and patients seeking healthcare other than for infertility were excluded.

### 2.2. Research Instrument

A structured questionnaire was developed to collect information on sociodemographic and lifestyle factors, fertility, and medical history through a self-administered questionnaire. The questionnaire consisted of mixed closed- and open-ended questions and was divided into four sections. The first section covered sociodemographic information including current age, age at marriage, gender, nationality, education, employment, income, and consanguinity. The second section covered lifestyle-related information including smoking (status, duration, frequency, and type), exposure to passive smoking, caffeine intake, physical activity, hot water/steam use, and potential exposure to radiation. The third section consisted of questions inquiring about medical-related characteristics including chronic comorbidities, history of genital infections and related symptoms, and intake of vitamins and nutritional supplements. The fourth section collected information on fertility-related characteristics such as age at puberty, menstrual cycle, number of pregnancies, method of achieving last pregnancy (spontaneous or with medical assistance), self-reported duration of infertility, number of children, history of miscarriage and ectopic pregnancy, and family history of infertility. Information obtained from patients’ medical records included height; weight; hemoglobin levels; thyroid function; vitamin D deficiency; serology testing for syphilis, hepatitis type B and type C; and medical- and fertility-related characteristics. Medical records were also used to confirm information that was self-reported in the survey questionnaire, particularly information related to medical and fertility-related characteristics. In the event of any discrepancy between self-reported and data documented in medical records, the data from medical records were used in the present analysis. The questions covered in the survey questionnaire were carefully reviewed by the research team members independently. After implementing the necessary corrections and updates, an additional review round was provided by an expert researcher. This first review round aimed to have a clear understanding of the readability and understandability of the questionnaire. Then, the questionnaire survey was piloted to 20 subjects. Minor necessary amendments and rewording were adapted accordingly. This second review round aimed at ensuring that the survey questionnaire was understandable and easily answerable by the targeted study population. The survey questionnaire and medical data were linked by a unique code. No identifiers that could disclose the patient’s identity were collected.

### 2.3. Recruitment and Data Collection Process

Two well-trained recruiters approached patients who were consecutively visiting the two participating fertility clinics. Identified eligible patients were provided with a comprehensive information sheet according to their preferred language (English or Arabic) and invited to participate in the study. The information sheet provided a full explanation of the study objectives, methodology, data to be collected, and confidentiality of their information. Patients voluntarily agreed to participate in this study signed informed consent and completed a self-administered questionnaire. If a companion was not present, illiterate patients and patients who struggled to use the online survey platform were provided with the needed help by the recruiters.

Infertility diagnosis was confirmed from the patient’s medical records, and patients with missing information on infertility diagnosis were excluded. The type of infertility (primary or secondary) was determined according to the responses to questions related to the number of pregnancies, and the method of achieving pregnancy (spontaneous or assisted pregnancy) in the survey. Primary infertility was defined as a complete lack of a spontaneous pregnancy or the achievement of one pregnancy through assisted reproductive treatment. Patients were diagnosed with secondary infertility if they were unable to achieve pregnancy after having at least one spontaneous pregnancy [27]. In the event of data extraction for both partners, to avoid duplicate count and overestimation of the measured outcome (type of infertility), data from only the female partner were considered. Prioritizing collecting data from the female partner is justified by the fact that there was a more thorough investigation and available data found in the female partner’s medical records as well as the in-clinic presence of the female partner to answer the survey questionnaire. Primary causes of infertility reported as ICD codes were obtained from patients’ medical records. Semen analysis was retrieved from medical records included males’ semen analysis of surveyed females.

### 2.4. Ethical Considerations

The study protocol was approved by the Abu Dhabi Health Research and Technology Committee Institutional Review Board (Ref: DOH/CVDC/2020/1191) and HealthPlus Research Ethics Committee (REC/2020/P13). The study was conducted in accordance with the Declaration of Helsinki and adhered to Good Clinical Practice guidelines.

### 2.5. Sample Size Calculation

The minimum sample size was approximated at 385 surveys. The following inputs: the optimum value for the proportion (0.5), 5.0% margin of error, and a 95% confidence interval, were used in the formula (n = (Z^2^ × P(1 − P))/e^2^). Taking into consideration the 15% anticipated refusal rate, the final minimum sample size was recalculated to 450 participants.

### 2.6. Data Management and Statistical Analysis

Collected survey data were merged with data obtained from medical records. Data were cleaned, coded, and recategorized accordingly. Age at the time of the survey was presented as a continuous variable. We also classified the study participants into three arbitrarily chosen 10-year age groups (19–30, 31–40, and 41–54 years). The choice of these categories was made a priori to explore the association of primary fertility across age groups. BMI, calculated as body weight in kilograms (kg) divided by the squared height in meters (m^2^), was presented as a continuous variable and also classified into four groups: underweight (<18.5 kg/m^2^), normal (18.5–24.9 kg/m^2^), overweight (25.0–29.9 kg/m^2^), and obese (≥30.0 kg/m^2^) [28]. The aim of categorizing these continuous variables was to investigate whether the measured outcomes were concentrated in specific subpopulation groups. The duration of infertility describes the self-reported period of being infertile before the patient sought infertility treatment at the clinic. However, in this study, infertility diagnosis was verified from the medical records.

Categorical data were reported as frequencies and percentages and compared in patients with primary and secondary infertility using the chi-square test. The normal distribution of continuous variables was assessed using Shapiro–Wilk tests. Abnormally distributed continuous data were reported as the median and interquartile range (IQR), in addition to the mean and standard deviation (SD) and compared using nonparametric Mann–Whitney U tests for independent samples. Normally distributed continuous variables were reported as mean ± standard deviation (SD) and compared by two-sample t-tests. To control for the variation in the measured characteristics, a multivariate logistic regression model was performed to produce the adjusted odds ratio (aOR). The multivariate model included all characteristics that showed a significant association (*p* ≤ 0.05) with the type of infertility in the bivariate analyses. The model included current age and age at marriage as continuous variables, and nationality, BMI, duration of the marriage, consanguinity, ex/current smoking, exposure to passive smoking, chronic comorbidity, genital infection, age at puberty, and duration of infertility as categorical variables. In the multivariable model, when a variable was included as a continuous variable, the categorical form of this variable was not included, and vice versa.

All *p*-values were two-tailed. IBM SPSS Statistics version 26.0 was used for all statistical analyses, with *p* values ≤ 0.05 defined as statistically significant.

## 3. Results

### 3.1. Sociodemographic Characteristics

During the study period, 1616 patients attending the two surveyed fertility clinics were invited to participate of which 1040 consented to participate. Patients recruited during the piloting phase were excluded from the analysis. The excluded patients from the pilot study had a distribution in their type of infertility, gender, and nationality comparable to that of patients included in the present analysis. After exclusion, 928 (89.2%) were included in the final analysis (Figure 1). The mean age (±SD) of the patients included in the analysis was 35.7 (±6.7) years with 55.0% aged ≥30 years. Of the included patients, 90.0% were women, 88.0% were of Middle Eastern nationality, 67.9% were college-educated, 57.2% were engaged in an income-generating job, and 71.8% were overweight or obese. The patients’ mean age at first marriage (±SD) age was 25.2 ± 6.3 years. Slightly more than one-quarter (26.6%) reported consanguineous marriage, with 74.5% of the latter married to a first cousin (Table 1).

### 3.2. Lifestyle and Medical Characteristics

Current (7.3%) and ex-smoking (5.3%) were reported by 12.5% of patients, with 60.0% of these individuals having been smokers for more than five years. In addition, 27.6% of these patients reported exposure to passive smoking. Minimal or no physical activity was reported by 40.8% of patients (Table 2). Moreover, 23.2% reported at least one existing chronic comorbidity, with diabetes mellitus being the most frequent (8.2%), followed by chronic respiratory illnesses (6.5%). Of these patients, 14.0% had a previous urinary tract infection, and 17.5% had a genital infection, either currently or during the previous 3 months, with 54.1% of the latter patients having experienced genital infections more than once. The prevalence of syphilis, hepatitis B, and hepatitis C was 0.7%, 0.4%, and 1.1%, respectively (Table 3). Approximately 40% of patients reported at least one genital symptom, with genital itching being the most common (17.6%) (Figure 2). In addition, 46.1% of these patients had hypothyroidism, 21.2% had anemia, and 30.9% had a Vitamin D deficiency (Table 3).

### 3.3. Fertility-Related Characteristics and Causes

The mean age at puberty of male and female patients was 14.5 (±1.7 SD) years and 13.4 (±1.7 SD) years, respectively. Of the women surveyed, 75.6% had regular menstrual cycles, 68.3% had cycle lengths ≤28 days, and 11.4% had undergone varicocele repair. History of at least one pregnancy loss was reported in 33.8% of the patients, and 8.1% had an ectopic pregnancy. In addition, 27.2% of patients had a history of infertility in their parents and/or siblings, and 55.9% sought infertility treatments after >12 months of having regular unprotected intercourse. Female infertility of unspecified/other origin was the most common cause of infertility, being present in 82.7%. Other causes of infertility in women included infertility associated with anovulation (3.3%), infertility of tubal origin (2.8%), infertility of uterine origin (2.1%), infertility of pituitary-hypothalamic origin (0.2%), and infertility of cervical or vaginal origin (0.1%) (Table 4). Secondary causes of infertility in women included ovarian cyst in 98 (10.6%) patients, ovarian dysfunction in 97 (10.5%), polycystic ovarian syndrome (PCOS) in 87 (9.4%), tubal factor in 66 (7.1%), diminished ovarian reserve in 50 (5.4%) and anovulation factors in 37 (4.0%). The male factor was found in 15% of infertile women (Table 4 and Figure 3).

### 3.4. Types of Infertility and Factors Associated with Primary Infertility

Of the patients, 62.5% had secondary infertility and 37.5% had primary infertility. The mean (SD) age of patients with primary and secondary infertility were 34.2 ± 7.3 years and 36.6 ± 6.1 years, respectively, a difference that was statistically significant (*p* < 0.001). The frequency of patients from the Middle East was higher in secondary infertility compared to primary infertility. A significant association was shown across BMI classes and the type of infertility (*p* = 0.009). Of the socioeconomic factors, a statistically significant association was observed in patients’ income across infertility types (*p* < 0.001). Patients with primary infertility had a higher mean age at marriage (mean age 26.7 ± 6.8 SD years) compared to patients with secondary (mean age: 24.3 ± 5.8 SD years) (*p* < 0.001). We observed a significantly lower prevalence of consanguineous marriage in patients with primary infertility (15.4%) compared to patients with secondary infertility (33.2%) (*p* < 0.001) (Table 1).

Of the lifestyle factors, both smoking and exposure to passive smoking were associated with the types of infertility (*p* = 0.049 and *p* = 0.016, respectively). With regard to the medical characteristics, diabetes mellitus was twice as common in patients with secondary compared to primary infertility (*p* < 0.011), while respiratory diseases were more prevalent in patients with primary infertility (*p* = 0.025). Age at puberty was higher in patients with primary than secondary infertility (*p* = 0.003). Patients who reported seeking infertility treatment after >12 months of unsuccessful conception were four times more likely to have primary infertility than patients who were infertile for <3 months (Table 1).

After controlling for the variation observed in the bivariate analyses, the odds of being diagnosed with primary infertility was reduced by 6.0% with each one-year increase in age (aOR: 0.94, 95% CI: 0.91–0.98, *p* = 0.001). Compared with patients from Middle Eastern countries, those of other nationalities (Asian, European, African, or American), on average, were nearly twice as likely to be diagnosed with primary infertility (aOR: 1.9, 95% CI: 1.1–3.3, *p* = 0.014). Compared to patients who reported being married for more than 5 years, the odds of being with primary infertility was higher in patients who had been married for ≤ 5 years (aOR: 6.0, 3.9–9.3, *p* < 0.001). Patients without any degree of consanguineous marriage were significantly more likely to be diagnosed with primary infertility (aOR: 2.4, 95% CI: 1.5–3.9, *p* < 0.001) compared to patients who reported a consanguineous marriage. Patients who reported having respiratory diseases were also at increased odds of being diagnosed with primary infertility (aOR: 2.3, 95% CI: 1.1–4.6, *p* < 0.001). Every one-year increase in age at puberty was associated with 20% increased odds of being with primary infertility in both genders (aOR: 1.2, 95% CI: 1.0–1.3, *p* = 0.006) and in females separately (aOR: 1.2, 95% CI: 1.1–1.3, *p* = 0.003), respectively. The odds of being diagnosed with primary infertility was 3.4-times and 2.4-times higher for patients with a self-reported duration of infertility for >12 months (aOR: 3.4, 95% CI: 1.8–6.4, *p* < 0.001) and for 6–<12 months (aOR: 2.4, 95% CI: 1.2–5.1, *p* = 0.016), respectively, compared to patients with a self-reported duration of infertility for <3 months. Patients diagnosed with infertility of anovulation, tubal, or uterine origin were also at higher odds of primary infertility (aOR, 3.9, 95% CI: 1.9–7.9, *p* < 0.001) compared to patients with unspecified causes of infertility (Table 5).

## 4. Discussion

This cross-sectional study evaluated the sociodemographic, lifestyle, and medical and clinical characteristics of patients who attended two major fertility clinics. This enabled identification of the most prevalent type of infertility, as well as factors associated with primary infertility. Secondary infertility was found to be 1.7 times more common than primary infertility in this study sample. This is consistent with experts’ opinions stating that the UAE is considered among the highest secondary infertility rates in the world [29]. It is consistent with, but lower than, estimates reported from several countries. For instance, a study in Oman, a neighboring country to UAE, revealed three-time higher prevalence of secondary infertility (71.5%) than primary infertility (28.5%) [30].The mean age of the surveyed fertility clinics’ attendees was relatively advanced (35.7 ± 6.7 years). Surveyed patients were generally of high socioeconomic status, with a mean age at marriage of 24.0 years, close to the mean age at marriage of Abu Dhabi female citizens in 2018 (24.6 years) [21]. The increase in age lowers the reproduction span, and may cause both a reduction in ovulation, and increased risk of reproductive system disorders [31]. Generally, a delay in marriage due to completing additional years of schooling and employment results in the postponement of childbearing, which reduces the reproduction span. These findings are in line with global forecast analysis, suggesting that the worldwide reduction in fertility is associated with these factors [32]. Therefore, advanced maternal age may therefore contribute to infertility in the UAE. The observed relatively advanced mean age of the studied sample is similar to the mean age of infertile patients reported in Qatar [33], a neighboring country to the UAE.

The observed high prevalence of overweight and obesity among our studied sample of infertile patients was anticipated since overweight and obesity are prevalent in the general population in the UAE. The prevalence of overweight in the present sample of infertile patients was half (33.5%) that of the UAE general population (67.9%) as reported in the UAE National Health Survey 2018 [34]. On the contrary, obesity (38.3%) was more frequent among our sample of infertile patients included in this study compared to the adult population in the UAE (27.8%) [34]. This finding is in agreement with a recent study in the USA that presented a U-shaped curve between BMI and infertility [35]. Moreover, consanguineous marriage, smoking, obesity, lack of physical activity, and diabetes are prevalent in the UAE [36,37]. Therefore, the observed relatively common prevalence of these infertility-associated factors in the surveyed fertility clinic attendees was not surprising.

The burden of smoking in the surveyed sample of infertile patients (7.3%) was lower than that reported in the general population in the UAE (9.1%) [38]. However, we did not observe a significant association between smoking and primary infertility in this sample of infertile patients; however, this does not exclude the role of smoking in fertility complications as all patients surveyed in our study were infertile and there were no fertile patients to be compared with. Smoking impacts various parts of the reproductive tract including ovaries and fallopian tubes, in addition to impairment in folliculogenesis and embryo implantation [39]. Supporting evidence in a meta-analysis found a 1.6-fold increased likelihood of infertility in smoking women compared to non-smokers (OR: 1.60, 95% CI: 1.34–1.91) [40]. Similarly, second-hand smoking women in Qatar were 2.4 times more likely to be infertile (aOR: 2.44, 95% CI: 1.26–4.73, *p* = 0.008) compared to non-smokers [33]. In addition, our findings identified the occurrence of comorbidities, including diabetes and thyroid dysfunction which are known to be related to infertility. Almost half of our sample of infertile patients with information on thyroid dysfunction had a history of hypothyroidism. Mechanistically, unbalanced thyroid hormone levels directly affect the menstrual cycle and ovulation thereby reducing fertility [41].

The observed higher prevalence of secondary infertility is an alarming observation. In a sense, patients with secondary infertility were able to have children before. Becoming infertile after being fertile could possibly be linked to exposure to post-marriage-acquired risk factors. On the other hand, assessing factors associated with primary compared to secondary infertility is crucial given its consequences of never being able to achieve pregnancy. In this study, there was a positive association between primary infertility and young ages, normal BMI, shorter marriage duration, self-reported duration of infertility of ≥6 months, and delayed age at puberty. These features indicate a hormonal imbalance. Hormonal imbalance leads to delayed puberty, irregular periods, and anovulation [42] and these are classical signs of polycystic ovaries (PCO) [43]. The severe form is polycystic ovarian syndrome (PCOS), and a typical PCOS patient will be very young, thin, and will have irregular periods with anovulatory menstrual cycles [44,45]. Subsequent evidence suggests that PCOS is the most common cause of the ovulatory disorder and oligo-anovulation is related to an increased risk of infertility [43,46]. Hence, any young patient presented with primary infertility and with delayed age at puberty, normal BMI, and shorter duration of the marriage should be investigated for PCO and treated accordingly. However, the higher odds of being with primary infertility in patients of nationalities other than the nationality of a Middle Eastern country is possibly a coincidental finding that needs further investigation. However, it was previously reported that there is a higher or equal prevalence of primary infertility to that of secondary infertility in patients from non-Middle Eastern countries [47,48].

The observed association between respiratory diseases and primary infertility could possibly be a coincidental finding that warrants further investigation. However, this finding supports previous reports showing an association between respiratory diseases and primary infertility [49]. A high incidence of varicocele, one of the most important causes of infertility, was noticed in patients with COPD [50]. On the other hand, the negative association between age and duration of marriage with primary infertility reflects a positive association between age and secondary infertility. This positive association is rational because having a second child usually occurs at an advanced age compared to planning the first pregnancy. A similar finding was reported in Oman [51] and Saudi Arabia [52]. Consistent with the evidence from other studies, overweight and obesity, irregular menstrual cycles, hormonal imbalance, and ovulation were risk factors for infertility when compared to fertile patients [53,54].

In our study, the observed negative association between overweight and primary infertility indicates a positive association between overweight and secondary infertility. Following the same pattern, the observed negative association between consanguinity and primary infertility indicates a positive association between consanguinity and secondary infertility to which consanguineous marriage was reported to be related [55]. This can also be explained by the fact that patients with secondary infertility are overweight (higher BMI) as they have given birth to children previously and may have gained weight over time. Similarly, although consanguineous marriages can cause early miscarriage [56] it is not the main reason for primary infertility. Another finding from our study was the positive association between increased age at menarche and primary infertility. This finding is in line with a previous study that documented a positive association between increased age at menarche and infertility and subfecundity [6]. Women with delayed age at menarche were reported to have lower levels of Anti-Müllerian hormone (AMH) [57]. The concentration of AMH in serum is a marker of ovarian reserve and low levels of AMH in serum suggested to be an indicator of a smaller ovarian reserve [57]. Moreover, a reasonable explanation for the observed longer duration of which patients sought infertility treatments in primary infertility could be related to their younger age and the belief in having sufficient time to conceive.

### 4.1. Limitations

This study has limitations to be considered when interpreting the presented findings. The cross-sectional design and a lack of a comparative group of fertile patients prevent a determination of cause-and-effect relationships between the studied characteristics and type of infertility. The recall and reporting biases are unavoidable given the self-reported nature of the survey questionnaire and the sensitive nature of the studied topic. In addition, this study did not assess the nutritional, hormonal, and psychological aspects of infertility. Limiting collecting data from only one emirate, the generalizability of the presented findings to the wider infertile patients in the seven Emirates in the UAE should be carefully exercised. The convenience sampling method implemented in the present study also imposes a limitation on the generalizability of the presented findings. However, this sampling method was implemented as the most appropriate low-cost and easy method for addressing the objectives of the present study. Implementing another sampling approach such as simple random or systematic sampling would require more logistics and listing large number of patients in the sampling frame.

The lack of a validated questionnaire collecting data from patients with infertility in the UAE or in neighboring countries should not impose a potential bias on the presented findings. In the present study, using data from the survey questionnaire was limited only to the data commonly collected in surveys (e.g., age, income, education). Although the survey questionnaire included questions on fertility and medical history, classifying patients according to the type of infertility and characterizing their medical- and fertility-related characteristics was solely based on the data extracted from medical records.

### 4.2. Strengths

To our knowledge, this is the first clinic-based cross-sectional study of infertility including a large sample of patients seeking infertility treatment. The strengths of this study include its setting in two large and main fertility clinics in two different areas of the Abu Dhabi Emirate. Moreover, the large sample size improved the precision of the results.

## 5. Conclusions

One in three infertile patients were diagnosed with primary infertility. Several preventable fertility-related risk factors including overweight, consanguineous marriage, smoking, and diabetes were found to be common among the fertility clinic attendees. Primary infertility was positively associated with younger age, shorter duration of the marriage, being from a non-Middle Eastern country, suffering from respiratory diseases, being presented with a self-reported duration of infertility of ≥6 months, delayed age at puberty, and infertility of uterine origin, while it was negatively associated with overweight and non-consanguineous marriage. The positive association of primary infertility with young ages, delayed age at puberty, shorter marriage duration, and normal BMI, indicates a hormonal imbalance that is associated with PCO and PCOS. Therefore, patients presenting with primary infertility and these characteristics should be investigated for PCO and should be treated accordingly. Moreover, these findings could contribute to the development of a risk score assessment model to identify individuals at risk of experiencing primary infertility.

## Figures and Tables

**Figure 1 ijerph-20-01692-f001:**
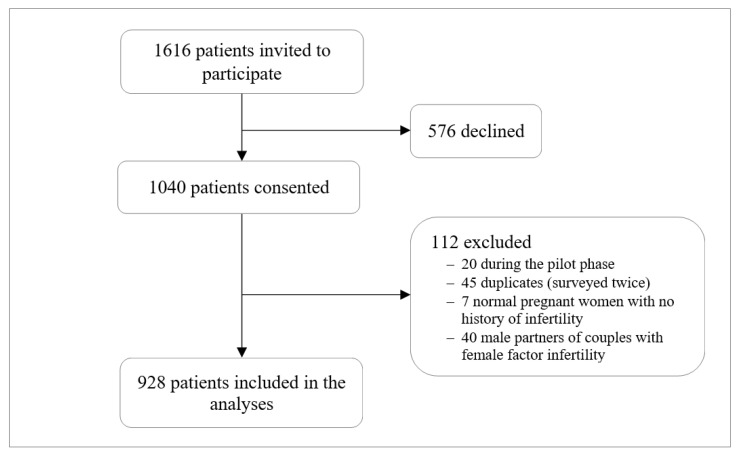
Flow chart of the surveyed fertility clinic attendees.

**Figure 2 ijerph-20-01692-f002:**
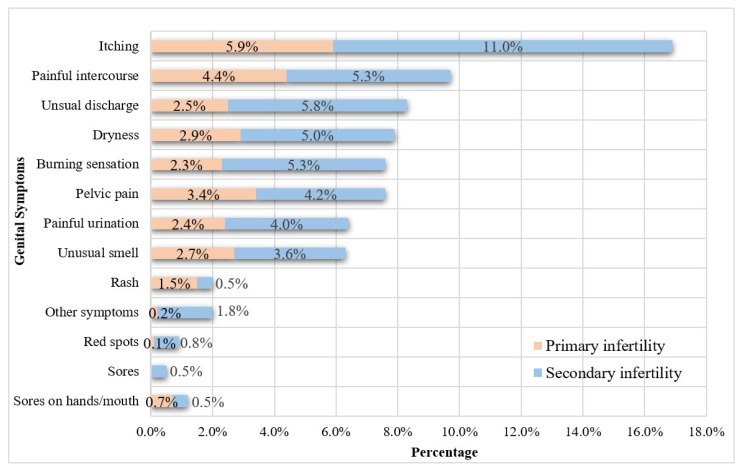
Genital symptoms reported by the fertility clinic attendees according to the type of infertility.

**Figure 3 ijerph-20-01692-f003:**
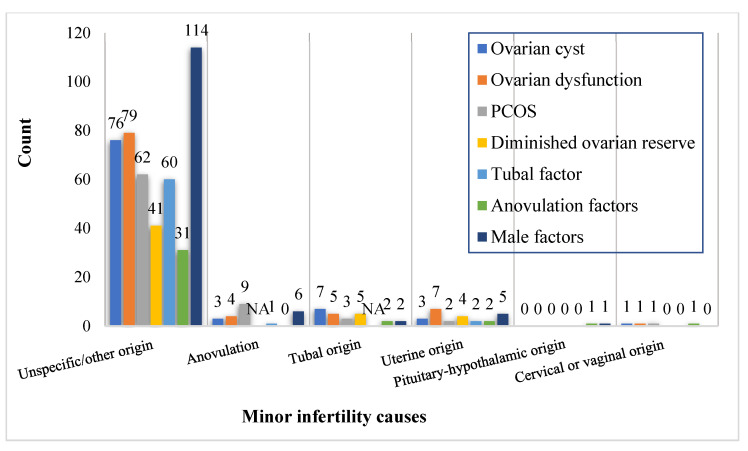
Distribution of fertility clinic attendees by the documented causes of infertility.

**Table 1 ijerph-20-01692-t001:** Sociodemographic characteristics of fertility clinic attendees, and the distribution between primary and secondary infertility.

	All N = 928	Infertility Type (n = 902)		*p*-Value *
		Primary n = 338 (37.5%)	Secondary n = 564 (62.5%)	
	n (Valid %)	n (Valid %)	n (Valid %)	
Age—years				
Mean ± SD	35.7 ± 6.7	34.2 ± 7.3	36.6 ± 6.1	0.001
41–54	253 (27.4)	76 (22.6)	169 (30.1)	0.001
31–40	448 (48.3)	142 (42.1)	293 (52.1)	
19–30	224 (24.2)	119 (35.3)	100 (17.8)	
Missing	3	1	2	
Gender				0.856
Male	93 (10.0)	33 (9.8)	53 (9.4)	
Female	835 (90.0)	305 (90.2)	511 (90.6)	
Nationality				0.001
Middle Eastern	803 (88.0)	272 (81.2)	515 (91.8)	
Others	110 (12.0)	63 (18.8)	46 (8.2)	
Asian	80 (8.8)	45 (13.4)	34 (6.1)	
European	13 (1.4)	7 (2.1)	6 (1.1)	
African	14 (1.5)	8 (2.4)	6 (1.1)	
American	3 (0.3)	3 (0.9)	0	
Missing	15	3	3	
Education				0.161
Secondary and below ^1^	290 (32.1)	98 (29.3)	189 (33.9)	
College level and higher	614 (67.9)	236 (70.7)	369 (66.1)	
Missing	24	4	6	
Employment status				0.424
Unemployed ^3^	390 (42.8)	139 (41.1)	246 (43.9)	
Employed/self-employed	521 (57.2)	199 (58.9)	315 (56.1)	
Missing	17	0	3	
Monthly income—AED				0.001
None	261 (32.6)	91 (30.2)	164 (33.5)	
<10 K	155 (19.4)	82 (27.2)	73 (14.9)	
10 K to <20 K	103 (12.9)	42 (14.0)	60 (12.2)	
≥20 K	282 (35.2)	86 (28.6)	193 (39.4)	
Missing	127	37	74	
BMI, kg/m^2^				0.007
Mean ± SD	28.8 ± 5.6	28.2 ± 5.7	29.2 ± 5.6	
Normal (18.5–24.9 kg/m^2^)	246 (27.3)	110 (34.0)	129 (23.4)	0.009
Overweight (25.0–29.9 kg/m^2^)	302 (33.5)	97 (29.9)	197 (35.8)	
Obese (≥30.0 kg/m^2^)	345 (38.3)	114 (35.2)	220 (39.9)	
Underweight (<18.5 kg/m^2^)	8 (0.9)	3 (0.9)	5 (0.9)	
Overweight and obese (≥25.0 kg/m^2^)	647 (71.8)	211 (65.1)	417 (75.7)	
Missing	27	14	13	
Age at marriage—year				0.001
Mean ± SD	25.2 ± 6.3	26.7 ± 6.8	24.3 ± 5.8	
Missing	43	7	16	
Marriage duration				0.001
<1 year	43 (4.7)	38 (11.2)	5 (0.9)	
1–5 years	294 (32.4)	180 (53.3)	114 (20.2)	
>5 years	570 (62.8)	120 (35.5)	444 (78.9)	
Missing	21	0	1	
Consanguinity occurrence				0.001
No	666 (73.4)	286 (84.6)	376 (66.8)	
Yes	241 (26.6)	52 (15.4)	187 (33.2)	
Missing	21	0	1	
Consanguinity degree (n = 241)				0.185
First cousins	76 (74.5)	14 (63.6)	62 (77.5)	
Second cousins	12 (11.8)	5 (22.7)	7 (8.8)	
First cousins once removed	10 (9.8)	3 (13.6)	7 (8.8)	
Relatives, unspecified	4 (3.9)	0 (0.0)	4 (5.0)	
Missing	139	30	107	

^1^ Five had no schooling, ^3^ including students and retired persons. * *p*-values obtained from the chi-square tests for categorical variables and from two-sample t-test for continuous variables. SD: standard deviation; AED: Arab Emirates Dirham.

**Table 2 ijerph-20-01692-t002:** Lifestyle characteristics of fertility clinic attendees, and the distribution between primary and secondary infertility.

	All N = 928	Infertility Type (n = 902)		*p*-Value *
		Primary n = 338 (37.5%)	Secondary n = 564 (62.5%)	
	n (Valid %)	n (Valid %)	n (Valid %)	
Smoking				0.049
Non-smoker	796 (87.5)	284 (84.8)	499 (89.3)	
Ex/current smoker	114 (12.5)	51 (15.2)	60 (10.7)	
Missing	18	3	5	
Duration of smoking ^1^ (n = 114)				0.342
<1 year	10 (9.1)	6 (12.0)	3 (5.2)	
1–5 years	34 (30.9)	13 (26.0)	20 (34.5)	
>5 years	66 (60.0)	31 (62.0)	35 (60.3)	
Missing	4	1	2	
Frequency of smoking ^1^ (n = 114)				0.514
Rarely	8 (7.2)	4 (7.8)	2 (3.5)	
Daily	69 (62.2)	29 (56.9)	39 (68.4)	
Weekly	23 (20.7)	13 (25.5)	10 (17.5)	
Monthly	11 (9.9)	5 (9.8)	6 (10.5)	
Missing	3	0	3	
Type of smoked tobacco ^1^ (n = 114)				NA
Cigarette	43 (37.7)	26 (63.4)	15 (36.6)	
Shisha	39 (34.2)	17 (44.7)	21 (55.3)	
Medwakh	27 (23.7)	10 (38.5)	16 (61.5)	
E-cigarette	27 (23.7)	9 (34.6)	17 (65.4)	
Passive Smoking				0.016
No	659 (72.4)	228 (67.9)	421 (75.3)	
Yes	251 (27.6)	108 (32.1)	138 (24.7)	
Missing	18	2	5	
Caffeine intake				0.567
None	109 (11.9)	44 (13.0)	65 (11.6)	
Occasional	155 (17.0)	53 (15.7)	100 (17.8)	
≤5 cups/day	625 (68.4)	235 (69.5)	380 (67.7)	
>5 cups/day	25 (2.7)	6 (1.8)	16 ((2.9)	
Missing	14	0	3	
Physical activity				0.570
Minimal or none	373 (40.8)	144 (42.6)	223 (39.8)	
1–5 times/week	476 (52.0)	173 (51.2)	295 (52.6)	
>5 times/week	66 (7.2)	21 (6.2)	43 (7.7)	
Missing	13	0	3	
Intensity of physical activity ^2^ (n = 542)				0.519
Light	99 (18.8)	31 (16.1)	67 (20.2)	
Moderate	281 (53.2)	105 (55.9)	171 (51.5)	
Intense	148 (28.0)	52 (27.7)	94 (28.3)	
Missing	13	0	3	
Hot water/steam use				0.782
No	592 (65.2)	220 (65.7)	362 (64.8)	
Yes	316 (34.8)	115 (34.3)	197 (35.2)	
Missing	20	3	5	
Exposure to radiation				0.412
No	869 (97.0)	320 (97.6)	536 (96.6)	
Yes	27 (3.0)	8 (2.4)	19 (3.4)	
Missing	32	10	9	

^1^ Including only current smokers and ex-smokers. ^2^ Excluding participants who reported rare or no exercise. * *p*-value obtained from the chi-square tests.

**Table 3 ijerph-20-01692-t003:** Medical characteristics of fertility clinic attendees, and the distribution between primary and secondary infertility.

	All N = 928	Infertility Type (n = 902)		*p*-Value *
		Primary n = 338 (37.5%)	Secondary n = 564 (62.5%)	
	n (Valid %)	n (Valid %)	n (Valid %)	
Chronic comorbidity				0.858
No	627 (76.8)	237 (76.2)	383 (76.8)	
At least one	192 (23.2)	74 (23.8)	116 (23.2)	
Missing	109	27	65	
Diabetes mellitus ^1^	67 (8.2)	16 (5.1)	51 (10.2)	0.011
Cardiovascular disorders ^2^	24 (2.9)	7 (2.3)	16 (3.2)	0.426
Respiratory diseases ^3^	53 (6.5)	28 (9.0)	25 (5.0)	0.025
Rheumatic arthritis	3 (0.4)	0 (0.0)	3 (0.6)	0.171
Thyroid dysfunction ^4^	19 (2.3)	8 (2.6)	10 (2.0)	0.594
Other chronic diseases ^5^	39 (4.8)	18 (5.8)	20 (4.0)	0.244
History of urinary tract infection				0.962
No	704 (86.0)	267 (85.9)	429 (86.0)	
Yes	115 (14.0)	44 (14.1)	70 (14.0)	
Missing	109	27	65	
History of genital infection				0.022
No	747 (82.2)	289 (86.0)	451 (80.0)	
Yes	162 (17.5)	47 (14.0)	113 (20.0)	
Missing	19	2	0	
Type of genital infection				0.211
Chlamydia	10 (6.2)	2 (4.3)	8 (7.1)	
Gonorrhea	1 (0.6)	0 (0.0)	1 (0.9)	
Trichomoniasis	5 (3.1)	2 (4.3)	3 (2.7)	
Herpes	3 (1.9)	0 (0.0)	3 (2.7)	
Other ^6^	23 (14.3)	11 (23.4)	11 (9.8)	
Unknown	119 (73.9)	32 (68.1)	83 (76.8)	
Missing	1	0	1	
Frequency of genital infection				0.743
Once	72 (45.9)	22 (47.8)	49 (45.0)	
More than once	85 (54.1)	24 (52.2)	60 (55.0)	
Missing	5	1	4	
Genital infection symptoms ^7^				0.616
None	445 (46.3)	173 (54.7)	269 (53.0)	
≥1	384 (53.7)	143 (45.3)	239 (47.0)	
Missing	99	22	56	
Syphilis (n = 545)	4 (0.7)	2 (0.4)	1 (0.2)	0.269
Hepatitis B (n = 548)	6 (1.1)	1 (0.2)	5 (1.0)	0.677
Hepatitis C (n = 536)	2 (0.4)	1 (0.2)	1 (0.2)	0.616
Dietary supplement use ^8^				
Vitamin D				0.818
No	299 (86.4)	119 (86.9)	178 (86.0)	
Yes	47 (13.6)	18 (13.1)	29 (14.0)	
Missing	582			
Iron supplements				0.368
No	329 (95.1)	132 (96.4)	195 (94.2)	
Yes	17 (4.9)	5 (3.6)	12 (5.8)	
Missing	582			
Folic acid supplement				0.026
No	276 (79.8)	101 (73.7)	173 (83.6)	
Yes	70 (20.2)	36 (26.3)	34 (16.4)	
Missing	582			
Steroid ^9^ (n = 93)				0.081
No	77 (89.5)	27 (81.8)	47 (94.0)	
Yes	9 (10.5)	6 (18.2)	3 (6.0)	
Missing	7	0	3	
Vitamin D deficiency ^10^	287 (30.9)	99 (29.3)	181 (32.1)	0.379
Anemia ^10^	197 (21.2)	70 (20.7)	123 (21.8)	0.697
Hemoglobin ^10,11^ (n = 676) (g/dL)				
Median (IQR),	12.2 (11.4–13.0)	12.6 (11.8–13.1)	12.1 (11.3–12.9)	<0.001
Mean ± SD	12.2 ± 1.3	12.4 ± 1.3	12.0 ± 1.3	
Male hemoglobin (n = 25)				0.970
Median (IQR)	15.2 (14.4–15.7)	15.3 (13.4–15.7)	15.0 (14.5–15.6)	
Mean ± SD	14.7 ± 1.7	14.3 ± 2.3	15.0 ± 0.7	
Female hemoglobin (n = 661)				<0.001
Median (IQR)	12.2 (11.4–13.0)	12.6 (11.7–13.1)	12.1 (11.2–12.8)	
Mean ± SD	12.1 ± 1.2	12.3 ± 1.2	12.0 ± 1.2	
Thyroid function ^10^ (n = 211)				0.207
Normal	61 (29.6)	23 (30.7)	38 (29.7)	
Hypothyroid	95 (46.1)	30 (40.0)	62 (48.4)	
Hyperthyroid	5 (2.4)	4 (5.3)	1 (0.8)	
Unspecified thyroid disorder	21 (10.2)	10 (13.3)	11 (8.6)	
Unspecified disorder of thyroid, pituitary, and hypothalamus	24 (11.7)	8 (10.7)	16 (12.5)	

^1^ Diabetes mellitus of any type including GDM. ^2^ Cardiovascular disorders of any type including high blood pressure. ^3^ Including COPD, asthma, and TB. ^4^ Thyroid dysfunctions of any type. ^5^ Cancer, allergy, anemia, PCOS, and PMDD. ^6^ Including bacterial (n = 4), viral (n = 1), and fungal (n = 3) infections, along with Candida (n = 2), thrush (n = 1), HPV (n = 1), and unspecified infection (n = 2). ^7^ Currently or during the previous three months. ^8^ Current medication use. ^9^ Sample size includes males only; ^10^ reported in medical records. ^11^ Nonnormally distributed variable (skewness: −0.448; kurtosis: 1.533; *p*-value obtained from Mann–Whitney U test). IQR: interquartile range. SD: standard deviation.

**Table 4 ijerph-20-01692-t004:** Fertility-related characteristics of fertility clinic attendees, and the distribution between primary and secondary infertility.

	All N = 928	Infertility Type (n = 902)		*p*-Value *
		Primary n = 338 (37.5%)	Secondary n = 564 (62.5%)	
	n (Valid %)	n (Valid %)	n (Valid %)	
Age at puberty—years				
Mean ± SD	13.5 ± 1.7	14.3 ± 1.6	14.6 ± 1.7	<0.001
Male age at puberty ^1^—years (n = 93)				
Mean ± SD	14.5 ± 1.7	14.3 ± 1.6	14.6 ± 1.7	
Missing	21	5	9	
Female age at puberty ^2^—years, (n = 835)				
Mean ± SD	13.4 ± 1.7	13.6 ± 1.7	13.2 ± 1.6	
Missing	48	13	23	
Menstrual cycle ^2^ (n = 835)				
Regularity				0.869
Regular	619 (75.6)	230 (75.9)	383 (75.4)	
Irregular	200 (24.4)	73 (24.1)	125 (24.6)	
Missing	16	2	3	
Cycle length				0.616
>28 days	258 (31.7)	99 (33.0)	159 (31.3)	
≤28 days	556 (68.3)	201 (67.0)	349 (68.7)	
Missing	21	5	3	
Number of days per period				<0.001
>5	439 (53.7)	138 (45.7)	296 (58.2)	
≤5	379 (46.3)	164 (54.3)	213 (41.8)	
Missing	17	3	2	
Varicocele repair ^1^ (n = 93)				0.421
No	78 (88.6)	28 (84.8)	48 (90.6)	
Yes	10 (11.4)	5 (15.2)	5 (9.4)	
Missing	5	0	0	
History of pregnancy loss				NA
No	644 (72.0)	332 (100.0)	306 (55.0)	
Yes	250 (28.0)	0	250 (45.0)	
Yes, once	129 (14.4)	0	129 (23.2)	
Yes, recurrent	121 (13.5)	0	121 (21.8)	
Missing	34	6	8	
History of ectopic pregnancy				NA
No	835 (93.4)	331 (100.0)	499 (89.6)	
Yes	59 (6.6)	0	58 (10.5)	
Yes, once	47 (5.3)	0	46 (8.3)	
Yes, recurrent	12 (1.3)	0	12 (2.2)	
Missing	34	7	7	
Family history of infertility (parents or siblings)				0.103
No	595 (66.6)	236 (70.9)	354 (63.9)	
Yes	243 (27.2)	79 (23.7)	164 (29.6)	
Unknown	55 (6.2)	18 (5.4)	36 (6.5)	
Missing	35	5	10	
Infertility duration ^3^				0.065
<3 months	144 (16.6)	43 (12.9)	100 (18.9)	
3–<6 months	123 (14.2)	49 (14.7)	74 (14.0)	
6–<12 months	116 (13.3)	53 (15.9)	62 (11.7)	
≥12 months	486 (55.9)	189 (56.6)	294 (55.5)	
≥6 months	602 (69.3)	242 (72.5)	356 (67.2)	
Missing	59	4	34	
Infertility causes ^4^ (n = 975)				0.006
Male infertility	71 (8.7)	23 (7.7)	43 (8.7)	
Other/unspecified Female infertility	675 (82.7)	241 (80.6)	417 (84.2)	
Female infertility associated with anovulation	27 (3.3)	9 (3.0)	18 (3.6)	
Female infertility of tubal origin	23 (2.8)	11 (3.7)	11 (2.2)	
Female infertility of uterine origin	17 (2.1)	14 (4.7)	3 (0.6)	
Female infertility of pituitary-hypothalamic origin	2 (0.2)	0	2 (0.4)	
Female infertility of cervical/vaginal origin	1 (0.1)	0	1 (0.2)	
Missing	131	40	69	
Semen analysis ^5^ (n = 100)				0.700
Normal	23 (23.0)	11 (21.2)	11	
Abnormal	77 (77.0)	41 (78.8)	43	

^1^ Sample size includes males only; ^2^ sample size includes females only. ^3^ Self-reported duration before the first visit seeking fertility treatment. ^4^ According to ICD coding, allows for overlapping when more than one cause reported on the same patient. ^5^ Includes surveyed males and surveyed females’ partners. SD: standard deviation. * *p*-value obtained from chi-square tests for categorical variables and from two-sample t-test for continuous variables.

**Table 5 ijerph-20-01692-t005:** Adjusted characteristics associated with primary infertility in fertility clinics attendees.

Characteristic	aOR * (95% CI)	*p*-Value
Age—year	0.94 (0.91–0.98)	<0.001
41–54	Reference	
31–40	1.4 (0.8–2.2)	0.247
19–30	2.8 (1.5–5.2)	0.001
Nationality		
Middle Eastern	Reference	
Others	1.9 (1.1–3.3)	0.014
BMI, continuous	1.00 (0.97–1.04)	
Normal	Reference	
Overweight	0.6 (0.4–0.9)	0.024
Obese	0.9 (0.6–1.5)	0.682
Underweight	0.5 (0.1–3.6)	0.464
Age at marriage—year	1.0 (1.0–1.1)	0.105
Marriage duration—year		
>5 years	Reference	
≤5 years	6.0 (3.9–9.3)	<0.001
Consanguinity occurrence		
Yes	Reference	
No	2.4 (1.5–3.9)	<0.001
Smoking		
Non-smoker	Reference	
Ex/current smoker	1.0 (0.5–1.7)	0.990
Passive Smoking		
No	Reference	
Yes	1.2 (0.8–1.8)	0.383
Chronic comorbidity (Yes vs No)		
Diabetes mellitus	0.7 (0.3–1.5)	0.360
Respiratory diseases	2.3 (1.1–4.6)	0.021
History of genital infection		
No	Reference	
Yes	0.7 (0.5–1.2)	0.201
Folic acid supplement use		
No	Reference	
Yes	1.4 (0.8–2.7)	0.316
Hemoglobin	1.1 (0.9–1.3)	0.195
Female hemoglobin	1.2 (1.0–1.4)	0.071
Age at puberty—year	1.2 (1.0–1.3)	0.006
Female age at puberty—year	1.2 (1.1–1.3)	0.003
Family history of infertility (parents or siblings)		
No	Reference	
Yes	0.7 (0.4–1.0)	0.068
Unknown	0.8 (0.3–1.8)	0.556
Infertility duration		
<3 months	Reference	
3–<6 months	1.5 (0.7–3.0)	0.333
6–<12 months	2.4 (1.2–5.1)	0.016
≥12 months	3.4 (1.8–6.4)	<0.001
Infertility causes		
Unspecified female infertility	Reference	
Female infertility of an ovulation, tubal, or uterine origin	3.9 (1.9–7.9)	<0.001

* aOR: adjusted odds ratio for age (as a continuous variable), nationality, BMI (as categorical variable), age at marriage (as a continuous variable), duration of the marriage, consanguinity, ex/current smoking, exposure to passive smoking, chronic comorbidity, genital infection, age at puberty, and duration of infertility. Goodness-of-fit: - 2LL = 711.413, Cox & Snell R^2^ = 0.279 Nagelkerke R^2^ = 0.379, Hosmer and Lemeshow test: *p* = 0.425.

## Data Availability

The data that support the findings of this study are available from authors, but restrictions apply to the availability of these data, which were used under license for the current study, and so are not publicly available. Data are, however, available from the authors upon reasonable request and with permission of the ethical approval provider.

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
