# Peer review of "Characterization of Fertility Clinic Attendees in the Abu Dhabi Emirate, United Arab Emirates: A Cross-Sectional Study"

_ijerph, 2023, doi:10.3390/ijerph20031692_

Round 1
Reviewer 1 Report
The authors describe the backgrounds of infertility patients in two clinics in UAE. The final number of patients analyzed was 928 and this work is very informative. However, some points needs to be addressed before publication.
- The authors describe that infertility according to WHO definition, not being able to conceive for over 12 months. The self-reported infertility period is shorter in many patients. Can you describe this more clearly in the text.
- In Table 4, infertility duration overlap with over 6 months and others. Why is over 6 months described here?
- I could not understand what the authors intended to describe in the discussion from lines 382 to 430. The paragraph is too long and facts are just stated all along.
- The p-values are italics in some tables and not others.
- Figure 3 is not a clear figure. Delete the horizontal lines and maybe use vertical lines to make the cause of infertility clearer.
- In table 3, at thyroid function, < before 0.207 is not needed.
Author Response
Response to Reviewer 1
Comments and Suggestions for Authors
The authors describe the backgrounds of infertility patients in two clinics in UAE. The final number of patients analyzed was 928 and this work is very informative. However, some points needs to be addressed before publication.
Response: We thank the reviewer for their time spent reading and evaluating our submitted manuscript. The authors have revised the manuscript in light of the insightful comments and suggestions provided by the reviewer. A point-by-point response is now made available addressing all comments and suggestions raised. Addressing those very important comments and suggestions have improved the readability and understandability of our work. All revisions to the manuscript are marked up using the “Track Changes” function. The revised manuscript is enclosed.
Comment #1. The authors describe that infertility according to WHO definition, not being able to conceive for over 12 months. The self-reported infertility period is shorter in many patients. Can you describe this more clearly in the text.
Response: Thank you for the very important comment. To avoid confusion, the authors would like to clarify that the self-reported infertility period is based on the period in which patients started seeking infertility treatment. We would also like to clarify that in the UAE, women aged >35 years unable to conceive after 6 months of regular unprotected sexual intercourse are considered as infertile patients. This has been already clarified in lines 102-103. Moreover, the authors have provided more elaboration on this point (please refer to lines 188 – 190). In our study, the diagnosis of infertility was verified from medical records.
Comment #2. In Table 4, infertility duration overlap with over 6 months and others. Why is over 6 months described here?
Response: Thank you for the very important comment. The authors would like to clarify that “≥6 months” is a separate subcategory combining the two subcategories ‘6 – 12 months and >12 months'. Referring to our response to the previous comment, this subcategory provides data on the proportion of patients who are over 35 years old and unable to conceive after 6 months of unprotected sexual intercourse, and who are defined in the UAE as infertile patients. To avoid confusion of overlapping between subcategories, the ‘duration of infertility’ variable is now fixed in Table 4.
Comment #3. I could not understand what the authors intended to describe in the discussion from lines 382 to 430. The paragraph is too long and facts are just stated all along.
Response. We thank the reviewer for their comment. In our study sample of patients, the indicated lines discuss the burden of the already-known risk factors for infertility that are documented in the literature. However, our discussion clarifies the fact the observed non-significant association between these or some of these factors with primary infertility does not exclude the role of those factors in infertility given the limitation that there was no ‘fertile’ group used as a comparison group to assess the risk of these factors with infertility. On the other hand, these lines, whenever applicable, also discuss that the factors that were negatively associated with primary infertility should be comprehended as factors positively associated with secondary infertility as our outcome variable is a binary variable of being with primary infertility coded with 1 or with secondary infertility coded with 0. The discussion in these lines is extended to show the importance of addressing these preventable factors in alleviating the burden of secondary infertility.
To avoid confusion and improve readability and understandability, these lines are now separated into several paragraphs (lines 370- 433).
Comment #4. The p-values are italics in some tables and not others.
Response. Thank you for the very important note. In all tables, the P-values are now made in the non-italic format.
Comment #5. Figure 3 is not a clear figure. Delete the horizontal lines and maybe use vertical lines to make the cause of infertility clearer.
Response. Thank you for the very important note. Figure 3 is now corrected as suggested by the reviewer where vertical lines are now provided in the figure to make the cause of infertility clearer.
Comment #6. In table 3, at thyroid function, < before 0.207 is not needed.
Response. Thank you for the very important note. The extra ‘<’ sign is now removed as it is not needed.

Reviewer 2 Report
Dear authors:
Thank you for submitting the current manuscript for consideration in the International Journal of Environmental Research and Public Health. The manuscript is sound, thorough and easy to read. I have nonetheless some concerns that should be addressed prior to considering the current version as a potential publication in the journal:
(1) Some English structures are awkward (for instance, line 96 " Participated clinics were"). The use of English should be improved throughout the manuscript;
(2) Sample size calculation: what was the margin of error chosen? All parameters should be stated;
(3) Line 191: Why the Shapiro–Wilk instead of Kolmogorov-Smirnov approach? Please justify;
(4) The fact of using convenience sampling highly limits the generalizability of your findings, and this fact should be highlighted beyond the limitations.
Author Response
Response to Reviewer 2
Comments and Suggestions for Authors
Thank you for submitting the current manuscript for consideration in the International Journal of Environmental Research and Public Health. The manuscript is sound, thorough and easy to read. I have nonetheless some concerns that should be addressed prior to considering the current version as a potential publication in the journal:
Response: We thank the reviewer for their time spent reading and assessing our submitted manuscript. The authors have revised the manuscript in light of the insightful comments and suggestions provided by the reviewer. A point-by-point response is now made available addressing all comments and suggestions raised. Addressing those very important comments and suggestions have improved the readability and understandability of our work. All revisions to the manuscript are marked up using the “Track Changes” function. The revised manuscript is enclosed.
Comment #1. Some English structures are awkward (for instance, line 96 " Participated clinics were"). The use of English should be improved throughout the manuscript;
Response. Thank you for the very important note. Line 96 is now corrected. The English in the revised manuscript was improved by a native English language speaker co-author.
Comment #2. Sample size calculation: what was the margin of error chosen? All parameters should be stated;
Response. Thank you for the very important note. The chosen margin of error was 5.0%. Now it is clearly provided in line #173.
Comment #3. Line 191: Why the Shapiro–Wilk instead of Kolmogorov-Smirnov approach? Please justify;
Response. Thank you for the very important note. The authors would like to clarify that they are aware that the Shapiro–Wilk test is more appropriate method for studies with sample sizes <50 subjects although it can also be handling on larger sample size while Kolmogorov–Smirnov test is used for studies with sample sizes ≥50. In our study, although both tests have provided same conclusion about the normality assumption, the authors opted to judge the normality assumption based on Shapiro–Wilk test as it does not assume that the population mean and variance are already known, which is the assumption of the Kolmogorov–Smirnov test.
Comment #4. The fact of using convenience sampling highly limits the generalizability of your findings, and this fact should be highlighted beyond the limitations.
Response. Thank you for this important comment. The authors have expanded the discussion to highlight the limitations of using convenience sampling beyond the generalizability of the presented findings. Please refer to lines 445 – 449.

Round 2
Reviewer 1 Report
The authors have answered all comments appropriately and the manuscript has improved.